# FORGIVE AND FORGET TO CREATE ROBUST, INTERPRETABLE MODELS

## ABSTRACT

Reaching internal transparency is a key challenge in the development of machine learning models. Rather than trying to interpret the models' internal structures, our approach aims to make that internal structure more interpretable. To this end, we introduce a trio of mechanisms that act on the FFNs of mT5 and the Channel Mixing layers of RWKV to produce similar outcomes: Proximal Forgetfulness, which considers weights spatially and forces them into clusters of similar magnitude; Forgiveness, which rewards close predictions to shape internal model structure and progression; and Fuzzy Recall, which shifts activations into related bands. In combination, these mechanisms are able to dramatically transform the models' internal topology in a controllable manner without compromising the performance of pretrained networks. Additionally, these changes make the model extremely resilient against noise and spatial perturbations. We show the modified internal topology is more dependent on the loss function than specific model architecture and can be crystallized if desired when changing tasks. With this new structure in place, internal token pathways can be represented with encouraging accuracy using a series of spatial centers and magnitudes. This is done without the use of a sparse autoencoder and could open the door to simplified control and interpretation in the future.

## 1 INTRODUCTION

Interpretability of transformer models is a critical area of focus in machine learning (Räuker et al., 2023; Sharkey et al., 2025). Much work has been done in recent years to develop a foundation of viewing these models' internal structure through a mechanical lens at various scopes, from the smallest neuron units (Dai et al., 2021) to larger attention head units (Olah et al., 2020; Olsson et al., 2022) and beyond. As the field of AI interpretability matured, there grew a clear need for a delineation between mechanical interpretability (the reverse-engineering of neural networks by understanding the causal role of mechanical units, such as weights, neurons, attention heads, circuits, etc) and the more abstract conceptual interpretability (the attempt to map internal representations, such as activations, latent spaces, clusters, etc) to human-meaningful concepts, with literature surveys and reviews created to examine the state of the field through these different lenses (Rai et al., 2024; Zhao et al., 2023).

Conceptual interpretability is the critical counterpart that we need to utilize our improved understanding of these models' mechanics in a useful way (Kim et al., 2018). For example, there is great interest in analyzing and controlling these models in areas as it pertains to machine learning safety (Gallegos et al., 2024), such as identifying bias conceptually (Kotek et al., 2023; Zhao et al., 2021) to adjust the models mechanically (Chandna et al., 2025; Yu & Ananiadou, 2025; Karvonen et al., 2025). In conceptual interpretability, particular interest is placed on the workings of the Feed Forward Layers of transformers (Geva et al., 2020; 2022). Discoveries of complex concepts distributed among activations of many sparse neurons in these layers (Gurnee et al., 2023) have spurred interest in the creation of neural activation decoding techniques (Foote et al., 2023; Zhao et al., 2024; Bills et al., 2023). Alongside these are other techniques that instead seek to create interpretability through reducing polysemanticity (Elhage et al., 2022), such as the Sparse Autoencoder (SAE) (Bricken et al., 2023; Templeton et al., 2024), a powerful tool for interpreting models, albeit with the drawback requiring large sample sizes for training stability (Huben et al., 2024). Our work seeks to take

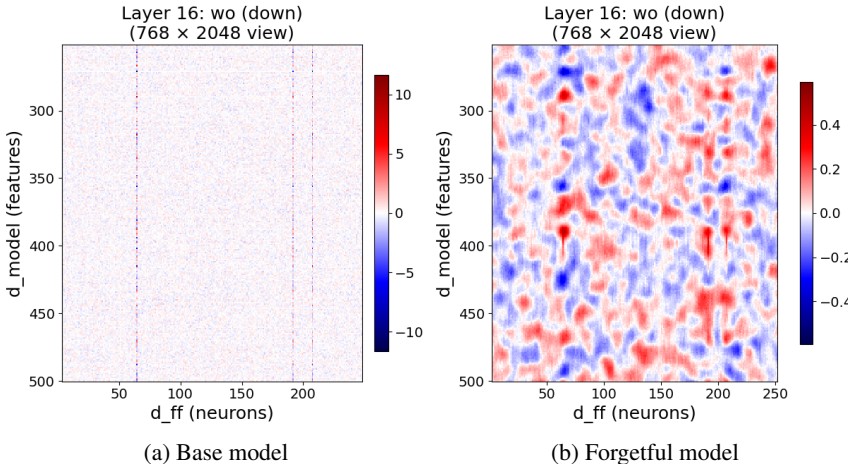

Figure 1: $200 \times 200$ slice of $W_o$ (layer 16). Full size $d_{\text{ff}} \times d_{\text{model}} = 768 \times 2048$.

steps toward a paradigm shift in interpretability: merging mechanical and conceptual interpretability through creating fundamental, integrated changes to network structure and activation patterns.

For this paper, we had a clear goal in mind: invoke changes in the network to shift its internals to make them easier to interpret. Besides basics like minimizing knowledge loss in the pretrained Multilingual Text-to-Text Transfer Transformer (mT5) (Xue et al., 2021) and Receptance Weighted Key Value (RWKV) (Peng et al., 2023) networks, we decided on a few core components that would be central to this task. First, establish a geographic representation for both weights and activations. Work by Li et al. (2024b) already show the potential of geometrically clustered representations in the networks of Sparse Autoencoders (SAE). Although the dimensionality of the system remains the same, if dimensions each other fire in similar ways it would make any actions done on them simpler. Second, create consistent clusters for related concepts and tokens and enable condense representation of those tokens within minimal clusters. Just ensuring activation and weight clusters does not mean the the contents of those clusters are totally related or that the clusters themselves are responsible for most expressivity. However, a model which did have this behavior would be much easier to map for storage and flow. If possible, we wanted to use this separability to map tokens in a simple lightweight way. Finally, we wanted to create some lever of control over this inner system forms. Generating related structured groupings could be useful by itself, but being able to determine their precise nature opens many more options.

## 2 CORE MECHANISMS

### 2.1 PROXIMAL FORGETFULNESS

Proximal forgetfulness (PFG) acts on the gate, value, and down projections weights in each Feed Forward Network (FFN) in mT5. At its core, this mechanism works by considering the location of these weights as spatially related. Since weights in these layers are centered around zero, we normalize after each pulse to prevent unintended weight decay.

In more precise terms, this utilizes a zero centered 2d kernel of size $k$ by $k$ and exponential decay $x$. This is then applied to each target weight by calculating the difference between the current weight and each weight in a distance based proportion based on the kernel. The Kernel size determines area of effect while decay shifts preference from near to far. Changing them will affect the exact structure of the clusters, but a wide range of values are stable. This kernel is then Exact equations for this and other mechanisms can be found in appendix C.

### 2.1.1 EFFECT ON WEIGHTS

The immediate effect of PFG is simple: the higher the difference in neighboring weights, the greater the change. If we consider weights as storage locations for how concepts relate to each other,

this mechanism makes it difficult to store very different memories near each other in the induced topology. Over time, this mechanism results in clusters of weights with similar magnitudes, as that is what minimizes the damage from each leak. Activations also group implicitly due to how they interact with the weights of each layer.

### 2.1.2 Training considerations

Since PFG permanently alters the core network structure, we make the power and frequency very delicate. On average, each weight receives a pulse every 200 batches with a maximum power equal to 3% of the difference in value to nodes in range.

When training a network from scratch, this mechanism and resulting structure could be implemented without worrying much about ramp time and with less than a 1% performance hit. However, we worked exclusively with pretrained networks in this paper. When fine tuning networks catastrophic forgetting is a significant concern (Luo et al., 2023), particularly in a situation like this where we are actively shifting weights (Li et al., 2024a). Because of this, we opted for a slow ramp over 10 million sentences which amounts to 70% of overall training time. This resulted in no measured drop off in performance, showing the model can forget its structure without forgetting its knowledge. It's likely more aggressive ramps are possible, but it is important to be mindful of the effects: our top end power applied without a ramp was enough to essentially wipe the network clean over just a five thousand sentences.

### 2.2 Fuzzy Recall

The clearly clustered weights and loosely clustered activations order PFG induces are significant. However, it is insufficient for our goal of crisp, interpretable internals. There are two main reasons for this: activations that flow through clusters are somewhat diffuse and the tokens that flow through them have a wide range of meaning even if they are more related than a base network. If PFG was applied from the start of scratch training it is possible that it would be enough to achieve cohesion with the loss function, as in theory, that would be the most efficient way to store information. However, for the timetable of fine tuning a pre-trained network, this is not the case.

When considering how to move the activation to more interpretable areas, there are a few options. A loss term could be added encouraging activation grouping like in the TopoLM language model (Rathi et al., 2025), but we opted to pursue a non loss option for reasons explained in section 2.3. A tilted spatial blur in activations could direct them to target areas, but that would force the information to essentially traverse the new weight topology. Since the weights are clustered by sign and magnitude at this point, underlying meaning of activations would have to be inverted several times to accomplish this, a very difficult task. However, there is a quirk in this topology: although the model does have to treat it as real geography if there are forces in place to make it that way, when designing the network we only have to use such rules when it remains beneficial.

That line of thinking resulted in the Fuzzy Recall (FZR) mechanism. Essentially, we provide the model with bird's-eye view information in the form of multiplicative noise. We first calculate 'hubs' at the token level during each forward pass, which are the areas in each layer with the strongest positive and negative activations using a uniform 1d kernel. Strong noise is applied outside these hubs, while noise inside is applied depending on how often they are used. This temporarily subverts the induced topology, allowing the model to quickly route information to far away regions while ignoring distance.

For this setup, the model's best defense against noise is redundancy (if a concept is represented cleanly by two nodes, taking the average essentially halves the noise) and activations that are clustered for a single token but diverse between tokens. This is also what gives us a window to influence the grouping, as reduced throughput gives the model heavy incentive to use the bird's eye information to group in a way that reduces the devastation of those effects.

### 2.2.1 Effects of Training

To examine the behavior of hubs, we trained with several different criteria before standardizing our approach. Across all training runs, one phenomena was constant: with a structured ramp, the model could withstand increasingly harsh noise with next to no performance loss until it reached a cliff. At

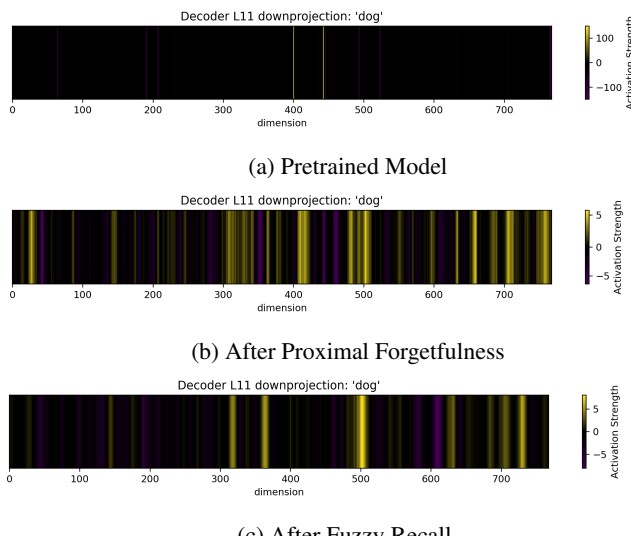

(a) Pretrained Model

(b) After Proximal Forgetfulness

(c) After Fuzzy Recall

Figure 2: Development of activation clusters by model training step. Over time, activation shift from infrequent spikes to bands with similar mass but smoothed distributions, then shift to concise, conceptually grouped clusters

that point, token level accuracy would quickly deteriorate to near zero. For conceptual forgiveness, we did one run with intra hub noise (branching model) and one without (highway model). This had a clear impact on hub usage: the most used nodes in the highway model were over twice as used as those of branching model. During the early stages of training, the easiest way for the model to adapt without adjusted hub noise is by pushing more of the activation mass through a few lanes. That is likely the cause of the disparity. This also had a large impact on the eventual collapse point; while the highway model collapsed at a range of 0-6x multiplicative noise, the branching model only reached that point when power reached a range of 0-12x. This collapse point was near identical even when we also halved the ramp speed after the 6x threshold. Together, this suggests two things. First, diversity of hubs allows for greater eventual performance. And second, the 'branches' of the hubs mostly settle in during the early stages of this phase. Intuitively, this makes sense: with more branches the model is less constrained by throughput when the task becomes hard, as there is too much information going through a few nodes to maintain both expressivity and redundancy. For the second point, it would be very risky for the model to try to branch out to new hubs once noise is already high, so it stays the course at the expense of future performance.

We also applied the full scheme to mT5 and RWKV models that were tuned on TLF but otherwise had identical settings. They both reached their collapse point at 0-7x multiplicative noise; a slight improvement over the highway model, but far below the 0-12x the main model was able to withstand. The less granular forgiveness (hundreds of 'close' answers for TLF compared to only 10 for conceptual) or an inefficient internal structure could be to blame for this. Although we did not prove the specifics, it does showcase a risk of exerting control of the models' inner architecture.

## 2.3 FORGIVENESS

Returning to our core task, PFG provides structure to the models' weights and activations. FZR makes those activations more focused within tokens and diverse between them and brings an element of conceptual coherence. Before diving into usage, one main facet remains: control. It can be useful to invoke structural changes on a network. However, without influence on how those clusters form, the ceiling and diversity of possible extensions remain limited. By manipulating the loss function, forgiveness provides that lever of control.

The idea behind forgiveness is simple: designate certain conditions where the model's incorrect predictions will be considered close, then reduce the loss the model receives proportionally. Withing that basic setup, there are endless opportunities to decide which tokens are close to each other and

how close they are. Since loss is what drives the core changes of a model, this simple mechanism is able to invoke a significant amount of control over the model's internals.

### 2.3.1 CONCEPTUAL VS TWO LETTER FORGIVENESS

To demonstrate its efficacy, we established two different forgiveness criteria. The first is conceptual forgiveness. For this, we train a teacher model that ranks tokens based on fit in the sentence. We then run the training dataset through the model with full information and save 10 alternatives for each token along with a score from the model indicating quality of fit. In practice, a very close token tends to be forgiven 60% of its loss in this setup while a loosely related one will be forgiven about 30%. This whole process (training and saving new dataset) only takes a few hours with our resources (see Appendix C). Data storage needs are significantly increased since the tokens are computed ahead of time, but, during runtime, vRAM and computation costs are both under 1%. Since the teacher model doesn't have to decipher masked tokens to see the full sentence, the task itself is much easier. Due to this, it is able to provide useful information for the workhorse models even with quick training. A good analogy would be a teacher with a thesaurus giving custom feedback to a student with no resources. Within that mold of using a weaker teacher with complete information to give feedback to a stronger student model with restricted information there could be many potential extensions adjacent to knowledge distillation (Hinton et al., 2015).

The second criteria is two-letter Forgiveness (TLF). This scheme gives half credit for tokens that begin with the same two letters as the target token. For example, 'pogo' and 'pound' would each receive half credit for the word 'pony.' This is not optimal for the task of span corruption, but that highlights the purpose: if clear groupings can be established in accordance with suboptimal criteria, it would be strong evidence that the loss function shapes the inner network structure.

### 2.3.2 ABILITY TO SHAPE ACTIVATIONS

As mentioned in section 2.2, this provides a lever to mold the internal topology generated by Proximal Forgetfulness and Fuzzy Recall. During the FZR stage in particular, bandwidth is limited and diversification is encouraged. As noise increases, throughput and potential avenues to diversify dry up. Thus, the model has a heavy incentive to form hubs that are related in the loss function, since any blending or misunderstanding would have smaller downstream consequences.

### 2.3.3 POTENTIAL FOR CLEANER GRADIENTS

We did not pursue this potential benefit in this paper as we wanted to focus on internal topology, but the possibility is clear. Similar to label smoothing or variants like Label Smoothing++ (LS++) (Chhabra et al., 2025), forgiveness has the opportunity to make cleaner gradients that speed learning. Intuitively, if a model is given a harsher penalty for something that is clearly very far (car vs truck vs Pam), it stands to reason that the resulting changes would be more accurate. However, this would only be true if the forgiveness criteria is supportive to the task at hand. Our first two letter forgiveness is a good example of this: somewhat rockier training and a collapse at 7x multiplicative noise compared to 12x for conceptual forgiveness suggest it worsened training to some extent. Unfortunately, showing that a bad criterion makes something worse does not suggest a good criterion must make things better, so the specific effects remain unknown for now.

## 3 USAGE OF NEW TOPOLOGY

### 3.1 ACTIVATION SIGNATURES

By using the combined training regiment described so far, we have access to spatially coherent groupings of activations and weights. However, the precision and separability of activations is a large contributor to how useful the signatures will eventually be. Expecting to isolate a token fully within one layer is unrealistic, but token 'signatures' that use several layers have much more viability (Katz & Belinkov, 2023; Ameisen et al., 2025).

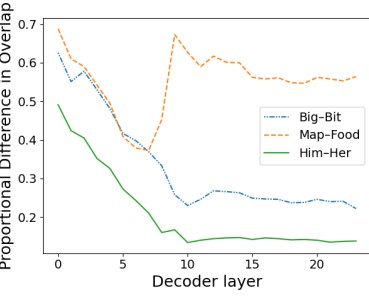

Figure 3: Overlap of different token pairs across layers in the RWKV model. The pattern of overlap maps varying significantly from layer to layer shows potential of inter layer separability.

### 3.1.1 IDEA AND ACCURACY

Table 1: Activation overlap across averaged instances of two tokens. Zero = complete overlap; one = no overlap.

<table>
<tr><td colspan="4" align="center">(a) Tok: 'bad'</td></tr>
<tr><td></td><td>Full Stack</td><td>TLF - FZR</td><td>TLF</td></tr>
<tr><td>Full Stack</td><td>0.157</td><td>0.268</td><td>0.414</td></tr>
<tr><td>TLF - FZR</td><td>–</td><td>0.181</td><td>0.393</td></tr>
<tr><td>TLF</td><td>–</td><td>–</td><td>0.137</td></tr>
</table>

<table>
<tr><td colspan="4" align="center">(b) Tok: '1'</td></tr>
<tr><td></td><td>Full Stack</td><td>TLF - FZR</td><td>TLF</td></tr>
<tr><td>Full Stack</td><td>0.067</td><td>0.303</td><td>0.342</td></tr>
<tr><td>TLF - FZR</td><td>–</td><td>0.099</td><td>0.322</td></tr>
<tr><td>TLF</td><td>–</td><td>–</td><td>0.086</td></tr>
</table>

To get a handle on the possible separability and uniqueness of token centers which are the basis of our initial token signature, we ran some diagnostics on the overlap of averaged token activations comparing models. When comparing a model to itself, two separate sets were averaged. We took the positive and negative activations of top 3% magnitude (with the full stack model that mostly lined up with hub centers) and took an overlap score where 0 indicates perfect overlap. There are several insights to be gained by examining these results. First, the overlap within a model is significantly higher than the training variants which indicates stable hub usage for a single token. This is highly impactful for the viability of signatures, as high separability between and correlation within hubs make many potential uses more feasible. Second, we should note that the TLF-FZR model was the base model with an extra 40% training time spent on TLF. The TLF model was trained with conceptual forgiveness the first 80% of training then ended with a TLF FZR combo. The fact that the full stack and TLF-FZR models are closer to each other than either are to the TLF model adds credence to the claim that conceptual groupings crystallize after FZR is turned off.

### 3.1.2 CRYSTALLIZATION OF SIGNATURES

Fuzzy Recall gives the model global information in the form of noise. Although most of the forces in this training regiment are set up to explicitly involve geography, FZR makes noise just as noisy regardless of distance, as long as they are outside of hubs. This not only gives an easier pathway to traverse the topology but also has the added benefit of a form of crystallization once that pressure is removed. With the weight structure as stable as it is, big geographic moves become very risky for the model. This causes the clusters to be stable even when the task changes.

To further isolate the source of conceptual grouping and show viability across architectures, we also compared activation overlap with two sets of tokens. One set was mostly random while the other was comprised entirely of pairs that start with the same two letters. Since each model has some differences in activations, we instead looked at the coverage trend between the two groups.

Table 2: Investigation into token grouping. For the Rand/2let line, a score of one would indicate no preference for grouping words that start with the same two letters over random tokens. Values greater than one demonstrate a bias towards two letter words while below one biases against them.

|  | Full Stack | TLF - FZR | TLF | RWKV TLF |
|---|---|---|---|---|
| 2let total | 0.690 | 0.850 | 0.807 | 0.567 |
| Rand total | 0.677 | 0.842 | 0.914 | 0.728 |
| Rand/2let | 0.980 | 0.990 | 1.130 | 1.280 |

In Table 2, a clear relationship surfaces among the models. The full stack and TLF-FZR models do not have notable bias either direction for TLF words. However, the TLF and RWKV TLF models both have a heavy bias towards grouping of TLF words. This provides strong evidence for three things: that loss is a driving force for token groupings, that groupings largely crystallize after FZR is switched off, and that loss is more important for conceptual groupings than the underlying model architecture. In short, it suggests that conceptual groupings in the induced topology materialize when the throughput of the network is constrained by FZR.

There are a few implications of this. First, it suggests that we can manipulate the makeup of the internal model architecture by influencing the loss function when certain conditions are met. Additionally, it demonstrates that FZR is likely what enables the smooth migration in the first place. For the model that ended on TLF but no FZR, in many ways the resulting internal architecture was less optimal than the TLF models we compared it to. The reason it did not shift to match them was because of inertia: despite some theoretical inefficiencies, the incentive to remove them did not exceed the security of continuing to look at things the same way. This opens a powerful pathway to train a model to form its structure in a way that is desirable on a task that is easy (like grouping similar concepts or behaving a certain way), then change to the true target without those bindings and still maintain most of the structure and interpretability. This seems particularly viable for pretrained networks that can then be fine tuned on numerous downstream tasks and remain structured without the overhead. From our results, we cannot conclude that the groupings persist over long times in all circumstances; it is likely some amount of drift is inevitable. However, the possibilities of this phenomenon are exciting even with that caveat.

### 3.1.3 USAGE AS A SIMPLE CLASSIFIER

To get an initial indicator on the performance of these signatures, we ran a simple classification algorithm (detailed in Appendix D) that matches activations to the closest known signature. We recorded a baseline for normalization, found signatures based on frequency for indices 51-150, then matched them to instances of the same tokens on a held out test set.

Table 3: Prediction accuracy on a group of 100 unique tokens using signatures. Random selection has an expected accuracy of 1%.

| Samples | Base Model | No FZR | Full Stack |
|---|---|---|---|
| 16 source, 1 target | 1.000% | 33.000% | 49.000% |
| 16 source, 16 target | 1.000% | 82.000% | 92.000% |

The results here are very encouraging. The base model scored 1% accuracy, which is expected with no spatial structure to support this method. The model without FZR performed admirably, but the addition of FZR improved both context invariance (single target sample) and stability (16 averaged samples) considerably. We should also note that on the experiment with 92% accuracy, half of the incorrect samples were deemed 'close' by the forgiveness metric. Considering that for each word only ten in the entire vocabulary have this designation, this happening at random is extremely unlikely. A more reasonably explanation is that this is a result of internal conceptual groupings.

More exploration is needed to uncover behavior for uncommon tokens or with an expanded vocabulary. However, for a non-learned classifier, these results are a promising start. Since signatures

are cross layer and have a massively reduced dimensionality compared to raw activations, there is clear potential here to simplifying the task of calculating distance and similarity between tokens and concepts.

# 4 GENERALIZATION AND ROBUSTNESS

## 4.1 RESILIENCE TO NOISE

Table 4: Collapse threshold for different types of noise

| Layer 23 | Base Model | No FZR | Full Stack |
|---|---|---|---|
| Activation Blur | 0.19 | 0.95 | 0.95 |
| Noise | 1.2 | 1.7 | 3.7 |
| Non-hub Noise | 2.3 | 4.7 | 18.6 |
| Non-hub Noise / Noise | 1.92 | 2.76 | 5.17 |

The addition of these mechanisms had a massive impact on resilience to various perturbations. To ensure validity, we also confirmed that FFN contributions to the residual stream remained stable across models. First off, for activation blur the proportion seen in Table 1 indicates the portion of difference in value from the current node and neighbors is applied at each step in the FFNs. Although a similar operation was applied to weights at various points, the extreme resistance to activation blur without directly changing it is indicative of the implicit effects on activations. This effect made the No FZR and Full Stack models over 10 times as resistant to activation blur as the standard model.

As for the multiplicative noise, based on the training regiment, it was almost guaranteed that the full stack model would increase resistance. Even though the full stack model was about three times as resistant as the base model, multiplicative noise like that doesn't occur in the wild so it is not very meaningful on its own. The boost does still suggests real redundancy gains as a result of exposure to noise in the full model and through some aspect of PFG in the non-FZR model which did not receive any noise. However, the bigger takeaway can be found by comparing the effects of total coverage noise and noise outside of hubs.

We only use one positive and one negative hub per activation for this test, each of which take up 3% of total area. This means only 3% of nodes are 'safe' for a node of a given sign. However, there is still a massive difference in how damaging the noise is if that small area is excluded, even for the standard model. This is not surprising, as if even a single node with super high activations is protected it could have significant downstream consequences. What makes this result influential is the proportional performance of non-hub vs everywhere noise across models. The full stack model is about three times as resilient to noise as the base model, but over eight times as resilient if hubs are excluded for both. This is strong evidence that the energy of a layer, or at least the important energy within a layer, has rerouted to go through centralized hub areas. This proportional gap is somewhat present in the no FZR model, but only the full stack completes the shift.

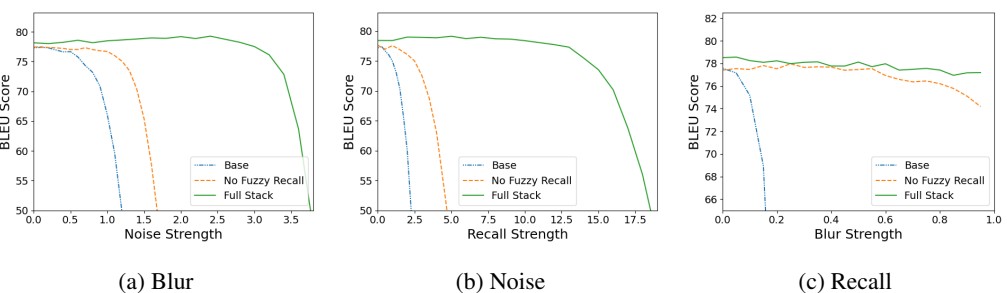

(a) Blur          (b) Noise          (c) Recall

Figure 4: BLEU vs. perturbation strength across models.

Table 5: Values of spatial metrics across models

| Layer 23 | Base Model | No FZR | Full Stack | Metric Range |
|----------|-----------|--------|-----------|--------------|
| Moran's I 2d | -0.0011 | 0.9449 | 0.9613 | [-1,1] |
| Moran's I 1d | -0.0047 | 0.9289 | 0.9627 | [-1,1] |
| LCPS 2d | 0.000 | 0.000 | 0.001 | [0,1] |
| LCPS 1d | 0.023 | 0.044 | 0.030 | [0,1] |
| RCI 2d | 0.0130 | 0.3925 | 0.3629 | [-1,1] |
| RCI 1d | 0.0625 | 0.4023 | 0.4590 | [-1,1] |

## 4.2 MORAN'S I

Moran's I is a measure of spatial autocorrelation (Moran, 1950), Ring Coherence Index (RCI) is a local coherence/edge contrast metric(Betzel & Bassett, 2017), and LISA Concentration Positive Score (LCPS) compares local and global clustering (Bivand & Wong, 2018). We used 1d and 2d variants to try to quantify the spatial effects from our mechanisms on activations and weights, respectively. To summarize, a Moran's I value of -1 indicates total anti correlation, 0 is random, and 1 is completely correlated. Higher values of RCI indicate sharper edges, while higher LCPS values indicate very skewed mass (positive and negative balance) across the measured areas. Overall, this gives us a very informed view of the resulting structure of our models. In simple terms, weights and activations are very similar over local neighborhoods (High Moran's I) but balanced globally (low LCPS). While boundaries are much more distinct than the base model, they are relatively smooth, not crisp (moderate RCI). Activations tend to be a touch crisper than weights once training is complete, possibly because FZR is does not directly smooth spatially like PFG.

## 4.3 INTERPRETATION

Taken together, this combination of robustness results is quite informative. The Moran's results show a strong grouping of weights and activations from PFG which become very strong once FZR is introduced. Notably, the change in smoothing from FZR is about twice the magnitude for activations as it is for weights. This establishes a related topology across activations and weights. Second, we observe an increasing importance of hubs demonstrated by the massively higher noise resistance on FZR models and somewhat higher on Forgetful models compared to baseline. This proportional difference would not be explained by a general increase in robustness; some amount of change in information routing is likely responsible. Finally, the activation smoothing test shows that the specific activation no longer matters as much as the base model. Although this would be expected to some extent just by grouping of activations, it is evidenced by the fact that the Full Stack model has about one third the performance loss at 0.95 blur as the forgetful model despite the main impetus for the change (PFG) remaining stable.

Although on their own these results would not be enough to conclude anything, taken together they all point in one direction: weights and activations have become spatially related and coordinated, routing similar information through correlated small areas while remaining balanced overall. Although not proven, the bias toward hub information, dwindling downstream effects for spatial blur on the Full Stack model, and high geographic correlation all point this way. If that is the case, it seems likely that concepts could be mapped across weights and activations to a large extent with this new network structure. Since weights and activations are grouped by magnitude and direction, each node in a cluster contributes to concepts presented there is more similar way than if activations were clustered but not the weights. That is one of the benefits of bringing structure to both weights and activations rather then one or the other as demonstrated by these results.

## 4.4 CROSS ARCHITECTURE

To show that these processes were not mT5 or transformer-specific, we trained a decoder only RWKV model. This model does not use attention or FFNs, but there are rough equivalents in the form of time mix and channel mix layers. We applied the same mix of mechanisms and hyperparameters as our base model but with TLF and achieved very similar outcomes. The main difference was that FZR began its collapse at 7x multiplicative noise compared to 12x for the base model. That

may be a symptom of inefficient grouping of TLF (the mT5 TLF model also collapsed at 7x noise) or it could be a coincidence; without knowing precisely what causes the collapse it is hard to tell. We did little more than compare the main tests for this model compared to the mT5 ones, so it is hard to say how much its behavior differs. However, it is clear the groupings were more similar between models with the same loss than models with the same architecture which is a promising result, and could be the first steps toward universal circuits (Chughtai et al., 2023).

## 5 DISCUSSION

While there certainly is more room for rigorous exploration of these mechanisms, our initial results of low performance degradation, signs of clustering promoting stable signatures, and novel functional pathways support the inclusion of our work among other contemporary interpretability methods. Our work differs from other frameworks like intrinsically interpretable models (Vandenhirtz et al., 2024; Kraus et al., 2024) and functional approximation (Ibrahim et al., 2023) in that we work directly within the weight/activation space of the FF layers. Our mechanisms function together without relying on structural constraints, such as the popular Concept Bottleneck methods (Koh et al., 2020), warranting our investigation, and confirmation, of comparable performance across architectures. Similarly, we avoid the need for context-specific surrogate models commonly found in methods focused on post-hoc explainability (Hakkoum et al., 2024). We treat topological smoothness as a first-class training signal and create decipherable geometries of weights and activations within the models' internal representations, hoping to achieve an elegant fusion of the explainability offered by leading dimensionality reduction methods (Wani et al., 2025) with the potential for human intervention and steering found in representation engineering methods (Lewandowski et al., 2024; Gao et al., 2025).

The viability of a topographically organized design philosophy in transformer layers producing functionally meaningful units was first demonstrated, through the context of cortical maps, by the Topographic Deep Artificial Neural Network TDANN (Margalit et al., 2024). It has since become a framework to create neural networks with regions of locally coherent functionality, inspiring models such as Topographic Vision Transformers (Shah & Yamins, 2025), Topographic Deep Spiking Neural Networks (Zhuang et al., 2025), and, most relevant to our work, Topographic Language Models TopoLM (Rathi et al., 2024). While TopoLM and our suite of mechanisms share similar end-goal design objectives and their target class of models, there are key differences, most notably that we demonstrate no major performance collapse when applying spatial regularization *after* the core pretraining. We retrofit our organization of weights and activations on top of a pretrained model, effectively nullifying the need for end-to-end training under spatial smoothness constraints. Furthermore, without having to specify a 2-D coordinate unit mapping at the beginning of training, as is generally required in TDANN inspired models, our method offers far more granular control over the final representation. This post-hoc flexibility in adapting the models' internal circuits into noise resistant, meaningful representations is a key consideration as we look toward creating networks that work *with* steering in large-scale language models where regular retraining is infeasible.

## 6 CONCLUSION

Through our set of mechanisms and structured training approach, we were able to invoke immense structural changes in two different model architectures while avoiding catastrophic forgetting. Activations and weights became highly spatially correlated with strong evidence showing that the loss function was the primary driver of grouping. When compared to the base model, we measured resistance to blur and multiplicative noise approaching ten times as high on our complete model at the same performance loss. By utilizing the tendency of tokens in this topology to be consistently routed through similar areas, we established a simple cross layer signature system with significant accuracy. Since we showed the state of FZR makes the loss-driven concept migration fluid or static, downstream applications for interpretable pretrained models seem much more plausible. Although we did not prove a direct locality, the rich relation between activations, concepts, and weights in the topology we induced merges aspects of mechanistic and conceptual interpretability. By reducing the difficulty of interpretability as a problem, there is reason to be optimistic that further innovations can sprout from this base.

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

## A    APPENDIX: EXTRA FIGURES

Table 6: Difference in Overlap for token pairs. 0 is perfect match, 1 is no overlap

|  | Full Stack | Base–TLF No FZR | TLF | RWKV TLF |
| --- | --- | --- | --- | --- |
| The & them | 0.315 | 0.386 | 0.374 | 0.234 |
| Bit & big | 0.364 | 0.464 | 0.433 | 0.333 |
| Map & food | 0.420 | 0.495 | 0.584 | 0.553 |
| Him & her | 0.257 | 0.347 | 0.330 | 0.175 |
| 2let total | 0.690 | 0.850 | 0.807 | 0.567 |
| Rand total | 0.677 | 0.842 | 0.914 | 0.728 |
| Rand/2let | 0.980 | 0.990 | 1.130 | 1.280 |

For the final row in the token pair table, a score of 1 indicates there is no preference for grouping based on the first two letters over random tokens. Values above one indicate a preference for grouping by two letter tokens while a value below 1 indicates an pressure against it. For each pair, an average of a few dozen instances were sampled.

## B    APPENDIX: EXPERIMENTAL DETAILS

Experimental details and standard hyper parameters can be found in attached python files. Batch size differed depending on training environment (most tests were run on a local 5090 but RWKV training and some two letter forgiveness training was run using online H100s) and sequence length for the RWKV model was increased to accommodate the decoder only architecture. Other parameters were generally equal to the ones currently set in the code. Key details like training ramp, kernel settings, forgiveness and noise settings were close to identical across the main training runs. The exception is for those dependent on batch size: to normalize the results, settings were adjusted so the frequency and power was consistent over the same number of sentences (eg, 2x batch size would result in twice as frequent proximal forgetfulness pulses and halved accumulation steps).

## C    APPENDIX: EQUATIONS

In our implementation of Proximal Forgetfulness, we use a decay of 1 and a kernel size of 9x9.

**Proximal Forgetfulness**

$$\tilde{M}_{ij} = M_{ij}\,\kappa\big(d(i,j)\big), \qquad\qquad \kappa(d) = \exp(-d/\tau) \qquad (1)$$

$$\mathcal{L}_{\text{pf}} = \lambda_{\text{pf}} \sum_{i \neq j} \big(1 - \kappa(d(i,j))\big)\,|M_{ij}|^p \qquad (2)$$

**Parameters:**

- $M \in \mathbb{R}^{n \times n}$: base inter-position weight/mixing matrix.
- $d(i,j) \in \mathbb{R}_{\geq 0}$: positional/graph distance between $i$ and $j$.
- $\tau > 0$: proximity length-scale (larger $\tau$ = less forgetting at distance).
- $\lambda_{\text{pf}} > 0$: regularization strength on distant connections.
- $p \in [1, 2]$: norm/exponent shaping the penalty on $M_{ij}$.

**Forgiveness**

$$\mathcal{L}_{\text{forg}} = \lambda_{\text{forg}} \sum_{i} \Big[ \max\big(0,\ \|z_i - \mu_{y_i}\| - m\big)\Big]^2 \qquad (3)$$

**Parameters:**

- $z_i \in \mathbb{R}^d$: latent representation of token/example $i$.
- $\mu_{y_i} \in \mathbb{R}^d$: prototype/centroid for class or type $y_i$.
- $m \geq 0$: forgiveness margin (zero loss if within $m$ of $\mu_{y_i}$).
- $\lambda_{\text{forg}} > 0$: weight of forgiveness objective.

**Fuzzy Recall**

$$\alpha_{im} = \frac{\exp\big(\langle z_i, h_m \rangle / \tau\big)}{\sum_n \exp\big(\langle z_i, h_n \rangle / \tau\big)}, \qquad\qquad r_i = \sum_m \alpha_{im}\, h_m \qquad (4)$$

$$\hat{z}_i = (1 - \gamma)\, z_i + \gamma\, r_i \qquad (5)$$

**Optional hub-shaping:**

$$\mathcal{L}_{\text{hub}} = -\lambda_{\text{ent}} \sum_i H(\alpha_i)\ +\ \lambda_{\text{mass}} \sum_m \Big( \sum_i \alpha_{im} \Big)^2 \qquad (6)$$

**Parameters:**

- $z_i \in \mathbb{R}^d$: current latent for $i$; $\hat{z}_i$ is the recalled/mixed latent.
- $h_m \in \mathbb{R}^d$: hub (codebook) vectors, $m = 1, \ldots, M$.
- $\tau > 0$: softmax temperature (lower $\tau$ = sharper recall).
- $\gamma \in [0, 1]$: mixing coefficient between direct state and recalled content.
- $\alpha_{im}$: recall weights (attention) over hubs; $H(\alpha_i)$ is entropy.
- $\lambda_{\text{ent}}, \lambda_{\text{mass}} \geq 0$: hub entropy and mass regularizers.

# D  APPENDIX: SIMPLE CLASSIFIER

After separating training and test sets, we found the 150 most common tokens in the training set. To avoid punctuation and other very common tokens, we then used numbers 51-150 to run our analysis.

To generate signatures, we found 16 instances of each of those tokens and averaged their 'hubs' at each layer. For a single layer, a signature consists of 1-3 positive hubs, 1-3 negative hubs, and a power associated with each. Each hub is represented by a single integer. A signature only has more than one hub if its power is at least 70% of the magnitude of the strongest hub.

Once source side signatures were compiled, we found the same samples on the held out test set. If the target samples were higher than 1, we averaged in the same way as before. Then, we performed a Champfer-style distance (Barrow et al., 1977) at each layer, then averaged that performance across all target layers. The token that had the lowest average distance was considered a match, and only that match was graded for performance (no partial credit). Overall accuracy as presented in the paper considered the actual matches vs possible matches in the 100 token test set.

In the 16 source, 16 target test which scored a 92% accuracy, half of the incorrect guesses were considered close by the forgiveness algorithm. They were as follows: '␣born, born. ␣South, ␣North. S, ␣S. ␣station, album.' These were also the four of the eight with the lowest distance. Basically, close by loss also indicated close by signature in this limited experiment. Since there are only 10 close tokens for each target token during training (which are context dependent), only 10% at most could be close in this target set. Since the vocab is many thousands of tokens, it is likely to be much less. Thus, the observed 50% close error rate holds some weight.

We also performed one final experiment comparing 250 tokens rather than 100. For the 16:16 setup on the full stack model, we measured an accuracy of 82.8%. Although this is a reduction in real terms, it is an improvement as a multiple of the random baseline (0.4% expected in this scenario) and makes the thought of scaling to a full vocabulary more promising.

# E  APPENDIX: LLM USAGE

We used LLMs to help source papers, write code, and decide between metrics. They were not used creating the theoretical designs of the various mechanisms, though they were occasionally used to validate. They were not used for paper writing besides referencing LaTex commands for figures and helping generate equations for the appendix.

