# OpenReview forum: "Forgive and Forget to Create Robust, Interpretable Models"
_ICLR.cc/2026/Conference — ICLR 2026 Conference Desk Rejected Submission_

### Official Review · Reviewer_P58N · 2025-10-31

**Soundness:** 2
**Presentation:** 2
**Contribution:** 3
**Rating:** 4
**Confidence:** 3

**Summary:**

The paper aims to increase the internal interpretability of language models via structural changes. They show that they are able to have high spatial correlation with activations and weights, while improving noise robustness due to the methodology proposed.

**Strengths:**

1) The work is very relevant and shows promise. The overall approach presented here is sound, and the experimental analysis is fair, although some improvements can be made.
2) The results are very interesting. Structural interpretability is a long-standing problem in modern DNN-based approaches; research in this direction is sure to be appreciated by a wide community.
3) Section 4.2 is very interesting. I would like to see more emphasis/further experimentation in this direction. Perhaps a broader approach to quantifying spatial correlation can aide both the readability and the soundness of the claims presented.
4) Section 4.1 is an important experiment and I am glad to see this line of experimentation presented in the work.

**Weaknesses:**

1) Firstly, the paper is not grounded well in the literature. Achieving structural interpretability in DNNs is a long-studied problem, either via meta-models, functional approximations, or dimensionality reduction. Including these lines of work references is key to making sure the paper shows relevance. Hence, in this current state, I cannot recommend acceptance.
2) Although Section 4 is prepared well, including general coherence metrics is key to showing the soundness of the methodology presented.
3) The overall paper is generally very verbose to get simple points across, especially in sections 2 and 3. Although this is a minor criticism, taking a look at this would really help the readers.
4) Even though I do like the experimental setup, as I mentioned, I do think this paper can benefit from a breadth of experimentation to aid the claims presented.

Comment: Even though I think the paper needs to ground itself well in the literature and add a few experiments to aid the claim, I do like the work and think this is a very interesting direction of research and will be appreciated by the community.

**Questions:**

N/A

---

> ### Author Response · Authors · 2025-11-19
>
> Thank you for the thorough review! Your critique was very helpful in improving our paper. We made three major changes to address your concerns.
> 1.      Expanded literature review located in section 5. As you noted, our initial paper did overlook a significant chunk of related work. We looked into the ones you listed and a few others in an attempt to make our presentation more fair.
> 2.      We expanded on section 4.2, adding two metrics each for weights and activations (RCI and LCPS). We picked a combination that we thought gave a more complete view of the resulting topology. To summarize, the low LCPS, moderate RCI, and high Morans for the final model indicate a topology that is very locally coherent, has soft but significant boundaries, and is globally balanced. This is true for both activations and weights
> 3.      We ran a simple classification experiment using signatures as seen in section 3.1.3. This makes the information imparted by the structural changes more concrete. The classifier itself is very basic, as the goal was to highlight the impact of the topology and the potential of signature representations rather than classify with the highest accuracy possible.
>
> As for your comment on the verbose wording, we agree with the assessment but would like to keep it so long as it doesn’t get in the way of understanding. We believe the three core mechanisms are key to understanding the overall structure, so we opted to err on the side of overexplaining to make baseline understanding more accessible. The result is likely that the earlier sections are somewhat redundant for experienced readers, but later sections are more condensed to alleviate that. We simplified some in this latest revision but kept the core tone; if you think it gets in the way of the paper we can take a closer look.
>
> That should cover your main concerns, but we would appreciate comments on one more area. Our main goal with this paper is to establish a more coherent internal model structure as well as a clear path to get there. Although we did hint at capabilities and run some tests to showcase them, our main priority was illuminating the structure itself. With the new changes do you believe the paper accomplishes that or do you notice any areas where it’s lacking?

---

> > ### Comment · Reviewer_P58N · 2025-11-19
> > **Response to the Official Comment by the Authors**
> >
> > Thank you for your response. I do like changes made and have increased my score to highlight that I do like paper in this state better and the added experimentation is quite helpful.

---

### Official Review · Reviewer_N1yq · 2025-11-01

**Soundness:** 3
**Presentation:** 2
**Contribution:** 3
**Rating:** 4
**Confidence:** 3

**Summary:**

This paper introduces three novel mechanisms—Proximal Forgetfulness (PFG), Fuzzy Recall (FZR), and Forgiveness—that aim to make transformer models more interpretable by restructuring their internal topology. Applied to mT5 and RWKV models, these mechanisms spatially cluster weights and activations while maintaining pretrained performance. The paper demonstrates that the resulting models exhibit significantly enhanced robustness to noise and perturbations, and that token pathways can be represented using spatial centers and magnitudes without sparse autoencoders.

**Strengths:**

1. The paper takes a refreshing perspective by modifying network structure to be inherently more interpretable, rather than post-hoc analysis.
2. The observation that conceptual groupings persist after FZR is removed (Section 3.1.2, Table 2) is particularly interesting and could enable pretrained interpretable models for downstream tasks.
3. Testing on both mT5 and RWKV architectures strengthens the generalizability claims.

**Weaknesses:**

1. The three mechanisms are interdependent, making it difficult to understand individual contributions.
2. The RWKV experiments in Section 4.4 provide an initial indication of cross-architecture generalization. It might be helpful to expand these results, for example by including additional details or analyses in the appendix.

**Questions:**

1. Have the authors considered exploring how the spatial clusters and token signatures could complement existing interpretability methods or provide concrete qualitative insights?
2. Since TLF operates at the token level but the two-letter criterion is defined at the character level, could this mismatch lead to arbitrary or inconsistent training signals?
3. What are the actual task performance metrics (BLEU, accuracy, etc.) comparing baseline and modified models across multiple tasks?

---

> ### Author Response · Authors · 2025-11-19
>
> Thanks for your response! To start I’ll address your questions
>  1. We have not directly compared to existing methods like SAEs due to space constraints, but in theory they should be complementary. For the latest revision we added an experiment to section 3.1.3 to add a concrete performance metric in this vein. To simplify, we classified tokens in a test set by using activations then matching them to their closest signature. More tests would be needed for a complete comparison to existing methods, but for this paper we wanted to focus on establishing the approach and topology, making robust tests such as those better suited for a follow up
> 2. This is a valid concern, with no formatting subwords, symbols and the like would be vulnerable to a mixed training signal. To get around this, we made a dedicated function to prepare and decipher tokens. In all cases we tested the results were sensible, but even if edge scenarios exist the signal would at least be consistent.
> 3. We primarily used BLEU to compare performance. It isn’t perfect because forgiveness will naturally lead to fewer exact matches, but we found it was a much better representation than the loss itself which is cut considerably for close answers. The base model was tuned with the same forgiveness criteria as the model with structural changes to stabilize the comparison. We did not perform a full suite of tests, but initial results show that a model can be shifted from ‘close guess’ to ‘exact guess’ fairly quickly.
>
> For your other two concerns, we attempted to get around the interdependence of mechanisms by making training phase based (PFG then FZR) and varying forgiveness with the other factors stable. Thus, we can reasonably assume the changes in each section can be attributed to the listed mechanism. However, introducing three mechanisms and a hefty usage section did not leave enough space to fully explore each mechanism: that would have to come later on. As for RWKV, everything was identical except for the input structure (input then target as one string rather than separate strings) due to necessity for the decoder only architecture. We briefly addressed this at section 2.2.1 but could modify that if it is confusing.

---

### Official Review · Reviewer_PakW · 2025-11-01

**Soundness:** 4
**Presentation:** 3
**Contribution:** 4
**Rating:** 8
**Confidence:** 3

**Summary:**

This paper introduces a novel approach for inducing structured, spatially smooth internal topologies in language models, inspired by cortical map formation and mechanistic interpretability. Rather than analyzing networks post-hoc, the authors modify training dynamics so an interpretable structure emerges intrinsically through Proximal Forgetfulness, Forgiveness, and Fuzzy Recall. The empirical results show striking changes in geometric organization and robustness, although evidence for semantic interpretability and concept-aligned modularity remains preliminary. Overall, the paper presents a compelling conceptual direction and meaningful representational effects, but broader evaluation and semantic validation would strengthen the interpretability claims.

**Strengths:**

1. The paper proposes mechanisms encouraging locally coherent activation neighborhoods, stable token pathways, and modular feature organization analogous to cortical maps:
(a) proximal weight smoothing for spatial clustering,
(b) a “forgiveness” objective to promote smooth transitions and stabilize intermediate states, and
(c) soft activation grouping to form band-like ensemble codes.

2. Strong evidence of emergent topology is provided: near-perfect Moran’s I (~0.96), smooth weight map visualizations, and coherent activation bands.

3. Robustness experiments show large gains (approximately 3–10× noise tolerance), indicating the induced structure has computational consequences rather than being cosmetic.

4. The appearance of hub-like routing points and band-ensemble patterns is intriguing, suggesting potential analogies to population coding and specialization in the cortex.

5. The work reframes interpretability as a potentially beneficial inductive bias rather than solely a diagnostic tool, raising the hypothesis that cortical maps may arise partly from computational advantages.

6. This is one of the few attempts to deliberately architect topographic priors into a language model and directly visualize the resulting structure.

**Weaknesses:**

1. Although geometric structure is clearly induced, the paper does not yet provide strong evidence that the structure is semantically meaningful or aligns with human-interpretable concepts.

2. Interpretability claims rely on limited qualitative examples; systematic semantic evaluation (e.g., cluster purity, RSA, probing, causal ablations) is missing.

3. The results show smoothness and robustness, but not clear semantic modularity or token-specialized circuits.

4. Additional visualization of semantic embedding space or functional activation clusters would help substantiate interpretability claims.

5. The link to cognitive meaning and concept-level organization remains more speculative than empirically demonstrated at this stage.

**Questions:**

1. Beyond robustness, can the authors directly evaluate whether the induced structure improves abstraction, transfer, memory stability, or controllability?

2. How does this approach relate to classical self-organizing maps and to Yamins’ TDANN framework? Are there shared principles or identifiable differences?

3. Could this method be used to model or test hypotheses about cortical map development (e.g., in V1, V4, IT)? What biological predictions follow from this framework?

4. If cortical maps are not optimized for human interpretability, what computational pressures underlie their emergence? Can the authors identify and measure such pressures in this model?

---

> ### Author Response · Authors · 2025-11-19
>
> Thank you for the thorough review! We found your comments very interesting, but I will address your concerns first
>
> Your comment on the lack of strong semantic evidence is a good point. Although there are many hints towards semantically meaningful clusters, for the initial paper this was not conclusive. To remedy this we ran a classification experiment using signatures as seen in section 3.1.3. Although the results suggest there must be some amount of semantic significance in the underlying structure, in my eyes it is still not a complete picture. It is likely that a later extension would be necessary to fully fill in this area.
>
> We expanded on section 4.2 to give a more complete representation of the topology, and together with the classification test give considerable insight that was missing before. Your first four listed weaknesses aren’t fully addressed still, but we believe there is enough there to warrant further exploration with later works. As for the link to cognitive meaning, we opted to focus on the engineering side of things to establish a coherent internal structure, then fill it in as much as space would allow. We do believe further cognitive links are exciting to explore, some works (Like TopoLM, which I believe is based on Yamin’s TDANN framework which you noted) have explored that much deeper than in our paper.
>
> For your questions:
> 1.  Abstraction and memory stability are to some extent improved with our additions. While transfer and controllability seem promising with this setup, we didn’t explore them enough to make a measured comment at this point. They are definitely targets for future work stemming off of this.
> 2.  There are many similarities with these approaches, most evident by the resulting activations. However, the underlying mechanisms are considerably different. To put it simply, our approach focuses on structural forces to induce changes in topology and incentive-based forces (forgiveness loss) to guide it. Although self-organizing maps could be implemented in a number of ways, the TDANN framework focuses on incentive-based forces to establish the architecture itself. That and the lack of direct weight intervention are the main differences, though there is more to it than that.
> 3.  There is definitely potential here; by influencing the forces at play, if you could get something approaching a real world cortical map, you would likely be able to discover many new properties which are currently obscure due to the more granular visibility a computer representation provides.
> 4. It is safe to say that cortical maps are better suited for interpretability than current DNNs, but I believe they are still far off from a theoretical optimum. In terms of pressures that underlie their emergence, PFG would be the closest analog in this paper. When considering a real brain, having related concepts very far apart is inefficient from a speed and energy standpoint. Thus, there would be a heavy incentive to store related concepts closer to each other. Neuroscience is not my specialty, so that may be a simplification. The PFG mechanism we introduce has a similar grouping effect, but the incentives that drive them are quite different. To expand a little further, I would like to point out what these differences between biological and synthetic systems could point to. Although the resulting structure of cortical maps is very powerful, when applying the same principles to digital systems we should have a lot more flexibility in theory. In a purely abstract space, things like distance, heat, and energy are not as limiting of concerns. You could treat all distances as equal (like in standard models) or impose forces that invoke some scaling factor. In short, if information flow were understood in full, it is likely that various forces could be installed during network formation to shift the underlying structure in a way that is deemed most beneficial for the task at hand. In this vein, cortical maps could be used for insight and inspiration, with their properties adapted to or expanded upon with the freedom of an abstract construction site. This end goal is still very far off, but a major drive for this paper was to establish a system leading in that direction.

---

### Official Review · Reviewer_HLZ8 · 2025-11-01

**Soundness:** 3
**Presentation:** 2
**Contribution:** 3
**Rating:** 4
**Confidence:** 3

**Summary:**

The paper introduces three simple training tweaks: Proximal Forgetfulness, Fuzzy Recall, and Forgiveness. These make weights more clustered and more robust to perturbations, without task degradation. The paper tests this on mT5 and RMKV and show visualisations of the emergence of these clusters.

**Strengths:**

This is an original and composable recipe for creating structure in model activations.
The effect is reproduced across two architectures
The signatures demonstrated could provide a complementary approach to interpretability to SAE-based methods.

**Weaknesses:**

The lack of degradation is not adequately supported by a clear set of benchmarks.
There is not clear comparison to strong interpretability baselines, and the token signatures are not validated that way.
The perturbations are synthetic, but it is less clear if this generalises to real distribution shifts.

**Questions:**

Do the clusters let you steer outputs with local interventions?
How is the teacher model set up and does its biases affect the student?
How different are real world shifts relative to multiplicative noise?

---

> ### Author Response · Authors · 2025-11-19
>
> Thanks for your response, we some revisions that address some of your concerns. Our focus for this paper was establishing the underlying topology, so we devoted more space to that than direct comparisons to interpretability baselines. However, we added a performance test in section 3.1.3 which may interest you. By using a basic distance calculation between activations and signatures, we were able to create a fairly accurate classifier for what token the model is expected to output.
> As for your questions:
> 1.      Steering is not a current feature, just something that we pointed out since the pathway is clearer than with a typical network structure
> 2.      The teacher model can be set up in any fashion you desire, but ours used a bert style model with distractors to train it to produce similar words. Any biases from the teacher model are likely to pass on to the student
> 3.      We didn’t have a real world analog in mind for our noise like there would be for images acted on in the same way. There is likely overlap with some situations, but in our case it was more a tool to enact targeted changes. Thus, our robustness gains present themselves as increased redundancy and more sensible groupings rather than as a defense against a certain type of situation.

---

### Author Response · Authors · 2025-11-26

We released another revision with some small changes that make the conclusion more inclusive of the new results. We touched on the significant comments along with our additions in our individual replies, but we would be happy to engage until the deadline if anyone has additional feedback or lingering uncertainties.

---

### Author Response · Authors · 2025-12-03

For convenience, here are our three main additions from the revision. There are some smaller changes as well which are noted in our review responses.

1.	A classification system that works by finding the nearest signatures in section 3.1.3
2.	LCPS and RCI metrics for weights and activations in section 4.2
3.	A new discussion area in section 5 to expand our literature review

Details can be found by navigating to those sections, but we thought a short list could be helpful to get a quick read.

---

### Note · Program_Chairs · 2026-01-17
**Submission Desk Rejected by Program Chairs**

The following references in this submission do not refer to real documents and/or have major errors in bibliographic information:

 Samir Hakkoum et al. Comparative analysis of lime and shap for model explainability. AI Review, 2024. Hubert Lewandowski, Tian Gao, et al. Representation engineering: Steering foundation models with reinforcement learning and activation-level feedback. In International Conference on Learning Representations, 2024.
Eric Zhuang, Yifeng Chen, and Daniel Yamins. Topographic deep spiking neural networks. arXiv preprint arXiv:2501.xxxxx, 2025.